# Convective heat transfer of the Taylor flow in a two-dimensional piston pump

**Liang Chang[1,2], Zhiwei Li[1,3], Wenang Jia[1], Sheng Li[1], Jian Ruan** [1] *

**1** Key Laboratory of Special Purpose Equipment and Advanced Manufacturing Technology, Ministry of Education & Zhejiang Province, Zhejiang University of Technology, Hangzhou, China, **2** School of Automobile, Zhejiang Institute of Communication, Hangzhou, China, **3** School of Transportation, Zhejiang Industry Polytechnic College, Shaoxing, China

* ruanjiane@zjut.edu.cn

**Data Availability Statement:** All relevant data are within the article and its Supporting Information files.

**Funding:** This research was funded by the National Key Research and Development Program of China,

## Abstract

Heat accumulation has become a key factor limiting the speed and pressure of pumps. Therefore, heat transfer analysis is an important and urgent task to analyze the mechanical efficiency and cooling performance. The derivation of the correct convective heat transfer coefficient is a basic work of calculating the accurate thermal state. This paper focuses on the Taylor flow heat transfer in a two-dimensional piston pump. Firstly, the theoretical and experimental studies are carried out on its thermal model to investigate the annular convective heat transfer coefficients, from 1000 rpm to 6000 rpm. Furthermore, the data are set in the transient thermal simulation model with Ansys software and the simulation results are mutually validated with the experimental ones. The Nusselt numbers are also calculated and compared with the empirical formulas. Two new Taylor flow relations, within 15% mean deviation, are deduced. As the rotational speed and oil temperature rise, the oil cavitation becomes more severe, restricting the convective heat transfer. Therefore, the thermal analysis must adopt the experimental ones rather than the empirical ones, above 3000 rpm. Finally, a modified relation is introduced to the gas-liquid two-phase flow heat transfer of the pump transmission.

## 1. Introduction

Pumps are at the heart of fluid power systems and become more intelligent, energy-efficient, and lightweight. To achieve a larger power/weight ratio, higher pressure and rotational speed are demanded by hydraulic pumps, especially for aerospace [1]. The even more enhanced pressure and power will inevitably increase the invalid power of the system, resulting in a profound rise in the temperature [2]. The pumps have to run at high temperatures, which cause efficiency decline, thermal stress fatigue, sealing problem, insufficient lubrication, working life reduction, etc. So the heat accumulation already becomes a key factor limiting the speed and pressure of the pumps.

Pump design techniques, efficiency, reliability, price, and operating conditions are researched by many groups and industries. The two-dimensional pump (2D pump) originates

grant number 2019YFB2005202, and by the Natural Science Foundation of Zhejiang Province, grant number LY21E050015, and by the Department of Education of Zhejiang Province, grant number Y201839694.

**Competing interests:** The authors have declared that no competing interests exist.

**Abbreviations:** a, Diameter ratio; $A_c$, Cross-section area of flow ($m^2$); $B_{i1}$, Biot number between the oil and inner wall; $B_{i2}$, Biot number between the air and outer wall; $c_{cy}$, Specific heat of the cylinder (kJ/kg·K); D, Shear rate (/s); $D_{h1}$, Hydraulic diameter of the inner annulus (m); $D_{h2}$, Hydraulic diameter of the outer wall (m); e, Annular gap thickness (m); Ein, Input energy rate (kJ/s); Eout, Energy output rate (kJ/s); Eg, Rate of energy generation (kJ/s); Est, Change rate of the cylinder's internal energy (kJ/s); G, Gap ratio; h, Maximum stroke of the piston (m); $h_1$, Convective heat transfer coefficient between oil and inner wall (W/m²·K); $h_2$, Convective heat transfer coefficient between surroundings and outer wall (W/m²·K) $k_{cy}$ Thermal conductivity of the cylinder (W/m·K); $k_1$, Thermal conductivity of the oil (W/m·K); $k_2$, Thermal conductivity of the air (W/m·K); $L_{cy}$, Annular fluid field length (m); n, Rotational speed (rpm); $N_{u1}$, Nusselt number of the inner wall; $N_{u2}$, Nusselt number of the outer wall; $P_{r2}$, Prandtl number of the outer wall; $q_{conv1}$, Heat-transfer rate of the oil and inner wall (kJ/s); $q_{conv2}$, Heat-transfer rate of the air and outer wall (kJ/s); $q_{rad2}$, Radiation heat-transfer rate of the outer wall and surroundings (kJ/s); $r_1$, Radius of the inner wall (m); $r_2$, the radius of the outer wall (m); $r_3$, Radius of inner annulus (m); $R_{a2}$, Rayleigh number of the outer wall; $R_e$, Reynolds number of the oil; t, Time (s); $T_a$, Taylor number of the oil; $T_{ac}$, Critical Taylor number of the oil; $T_{cy}$, Cylinder temperature (°C); $T_{f1}$, Film temperature of boundary layer 1 (°C); $T_{f2}$, Film temperature of boundary layer 2 (°C); $T_{No.1}$, Temperature of No.1 thermocouple (°C); $T_{No.2}$, Temperature of No.2 thermocouple (°C); $T_{No.3}$, Temperature of No.3 thermocouple (°C); $T_{No.4}$, Temperature of No.4 thermocouple (°C); $T_{oil}$, Oil temperature (°C); $T_{r1}$, Inner wall temperature (°C); $T_{r2}$, Outer wall temperature (°C); $T_{r2s}$, Simulation value of the outer wall temperature (°C); $T_\infty$, Environment temperature (°C); $T_{sur}$, Temperature of surroundings (°C); $V_{cy}$, Volume of the cylinder ($m^3$); $v_1$, Inner annular tangential velocity (m/s); $v_2$, Axial velocity of the main carrier (m/s); $v_2$', Axial velocity of the balancing carrier (m/s); $v_{2ave}$, Mean axial velocity of the carrier (m/s); $v_e$, Effective velocity of the carrier (m/s); Greek Symbols $\varepsilon$, Emissivity; $\theta$, Kinematic viscosity ($m^2$/s); $\mu$, Dynamicviscosity(N·s/m²); $\rho_{cy}$, Density of the

from the two-dimensional hydraulic component theory, which has been developed for over 20 years [3]. Ruan et al. [4] invented this kind of pump which is suitable for high power density applications, especially for aero applications. Multiple types of research have been carried out on the 2D pump efficiency, which is a critical characteristic. Shentu et al. [5] studied the flow characteristic of this pump. Huang et al. [6–8] pursue their research on energy losses. Zhang et al. [9] studied the volume efficiency of a stacked roller 2D pump. They have systematically analyzed its mechanical efficiency, and its energy losses can be carried away in the form of heat and vibration [10]. Although the mechanical efficiency has been analyzed theoretically and experimentally, the thermal analysis of this pump has not been studied yet. Heat loss analysis is important to analyze the mechanical efficiency.

The heat generating of this pump is due to two main aspects: the churning and compression of oil and the friction of mechanical parts. Zhang et al. from Zhejiang University [11] studied the churning loss analysis on the traditional axial piston pump. Huang et al. from Zhejiang University of Technology [12] analyzed the churning loss of the 2D pump's transmission. Qu et al. [13] from Maha Fluid Power Research Center utilized a lumped-parameter thermal model for electro-hydraulic actuators. Li et al. [14] from Beihang University carried out the thermal-hydraulic component simulation.

The convective heat transfer coefficients, between the oil and the pump shell, are the core elements to influence the heat transfer, cooling, and temperature prediction [15] of the 2D piston pump. Getting the correct convective heat transfer coefficients is important for the accurate calculation of the pump's thermal status. However, the convective coefficients differ particularly from the traditional regular ones, owing to the pump's unique structure and moving rules(shown in Fig 4). In its end cap, the roller sets, driven to rotate and reciprocate synchronously(shown in Fig 3), could be treated as an inner annulus. And the cap belongs to a stationary outer annulus [16]. The cap and the roller sets constitute a concentric cylindrical annulus with rotation of the inner cylinder, as shown in Fig 1. So the flow dynamics between two annuli are in the term Taylor-Couette flow [17]. Furthermore, such flow also belongs to the Taylor-Couette-Poiseuille flow, for certain axial flows existing [18]. And the convective heat transfer is from both annuli to the annular flow [19].

Since Couette and Taylor investigated the Taylor-Couette flow, the convective heat transfer in annuli has been a focus of intense research for years, particularly on the Nusselt number correlations, and numerous industrial applications exist [18]. Lots of variables affect the heat transfer: geometric parameters such as the annular diameter ratio, fluid field length, and flow variables such as Prandtl number, Reynolds number, and Taylor number (for rotational flow) [16]. The heat transfer coefficient is mainly influenced by the rotational Reynolds number or the Taylor number to the Taylor-Couette flow [20], and the additional axial flow rate coefficient to the Taylor-Couette-Poiseuille flow [21].

In 1958, Gazley [22] was the first author to show a sustained interest in the thermal study of Taylor-Couette flow. For the fully developed turbulent flow in the smooth tube, a well-known relation recommended by Dittus and Boelter [23] (for cooling of the fluid) is

$$N_u = 0.023 R_e^{0.8} P_r^{0.3} \tag{1}$$

And McAdams [24] suggested the following equation:

$$N_u = 0.03105 a^{0.15}(a-1)^{0.2} R_e^{0.8} P_r^{0.3} \left(\frac{\mu_b}{\mu_w}\right)^{0.14} \tag{2}$$

where a is the annular diameter ratio, $\mu_b$ and $\mu_w$ are the dynamic viscosity evaluated at bulk temperature and wall temperature. Both Eq 1 and Eq 2 are applicable to these concentric

cylinder (kg/m$^3$); $\Gamma$, Axial ratio; $\sigma$, Stefan-Boltzmann constant.

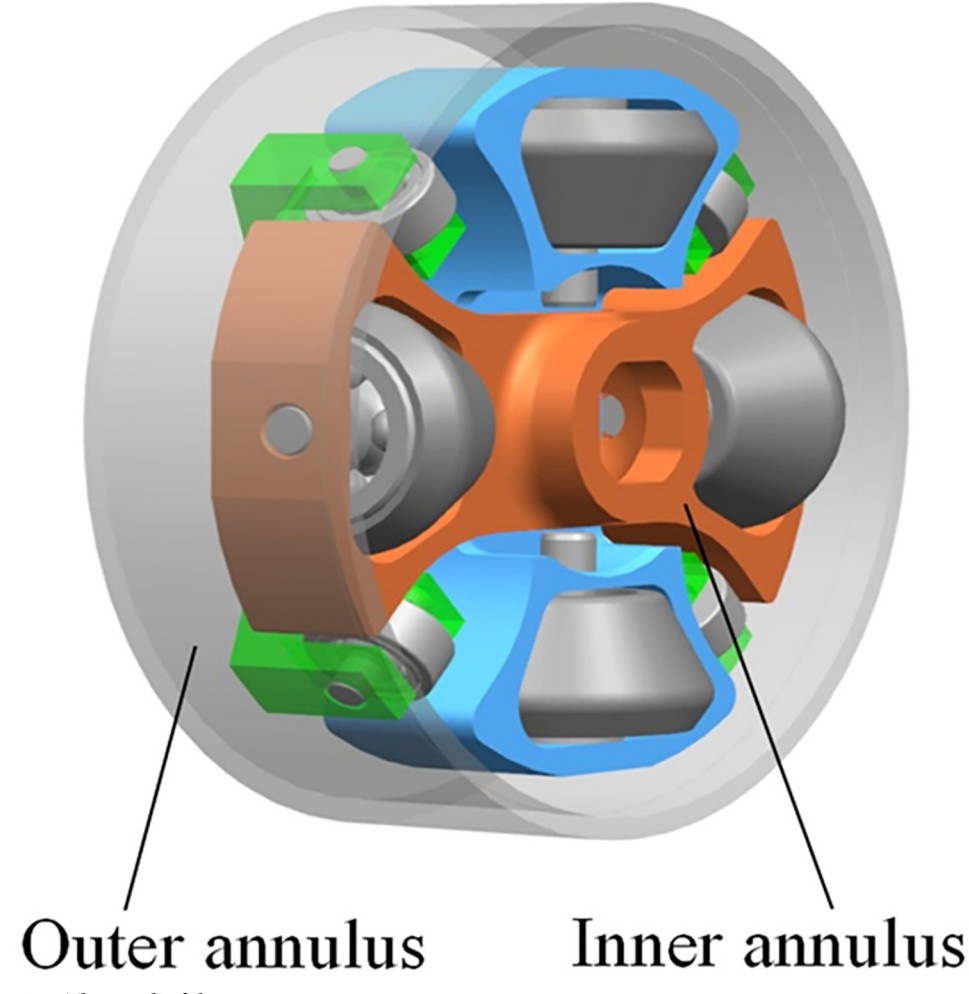

**Fig 1. The annuli of the 2D pump.**

annular ducts, in which diameter ratio range, Reynolds number range, and medium are all not specified [25].

Petukhov [26] developed a comparable acurate but complex relation:

$$N_u = \frac{(f/8)R_e P_r}{1.07 + 12.7(f/8)^{1/2}(P_r^{2/3}-1)}\left(\frac{\mu_b}{\mu_w}\right)^n \tag{3}$$

where $f = (1.82 \log_{10} R_e$-1.64$)^{-2}$, n = 0.11 for $T_w < T_b$, n = 0.25 for $T_b < T_w$. This equation is applicable for the ranges: $0.5 < P_r < 2000$, $10^4 < R_e < 5 \times 10^6$, and $0.8 < \mu_b/\mu_w < 40$.

Gnielinski [19] advises a more complicated expression, which may make the calculation error below 10%, for fully developed annular turbulent flow:

$$N_u = \frac{(f_{ann}/8)(R_e-1000)P_r}{1 + 12.7(f_{ann}/8)^{1/2}(P_r^{2/3}-1)}\left[1 + \left(\frac{D_h}{L}\right)^{2/3}\right]F_{ann}K \tag{4}$$

where $f_{ann} = (1.8 \log_{10}R_e^*-1.5)^{-2}$, $R_e^* = \frac{(1+a^2)\ln a + (1-a^2)}{(1-a^2)\ln a}R_e$, a is the annular diameter ratio, L is the annular length, $D_h$ is hydraulic diameter, $F_{ann} = 0.75a^{-2}$ for convection at the inner annulus with the outer annulus insulated, $F_{ann} = 0.9-0.15a^{0.6}$ for convection at the outer annulus with

the inner annulus insulated, $K = (P_{rb}/P_{rw})^{0.11}$ for liquids, $P_{rb}$ and $P_{rw}$ are the Prandtl numbers evaluated at bulk-temperature and wall temperature. This relation is applied for the ranges: $0.1 \leq P_r \leq 1000$, $R_e > 4000$ and $D_h/L \leq 1$.

Below the critical Taylor number, Tachibana et al. [27] and most of the other authors have concurrently found a constant value equal to 1. Becker and Kaye [28] studied the heat generated in rotating electrical machines with an air gap, and found $N_u = 2$ for laminar flow, by using the definition of a modified Taylor number $T_{am} = (\pi^4/1697)(1-e/2r_m)^{-2}/P$, $T_{am} = T_a/F_g$, where $F_g = (\pi^4/1697)(1-e/2r_m)^{-2}/P$, $e$ is the annular gap thickness, $r_m$ is the arithmetic mean radius of annulus, and $P = 0.0571 \left[ 1 - 0.652 \left( \frac{e/2r_m}{1-e/2r_m} \right) \right] + 0.00056 \left[ 1 - 0.652 \left( \frac{e/2r_m}{1-e/2r_m} \right) \right]^{-1}$. Besides they summarized a relation for $1700 \leq T_{am} \leq 10^7$:

$$N_u = 0.409(T_a/F_g)^{0.241} - 137(T_a/F_g)^{-0.75} \tag{5}$$

Tachibana and Fukui [29] have recorded an empirical equation without axial flow:

$$N_u = 0.21(T_a \cdot P_r)^{1/4} \tag{6}$$

where $T_a$ is below $10^8$ for air gap; and

$$N_u = 0.046(T_a \cdot P_r)^{1/3} \tag{7}$$

where $T_a$ is from $4 \times 10^8$ to $3.6 \times 10^{11}$ for the water gap. And they found the heat transfer coefficients of both sides heated are about 15% higher than that of one side heated and the other side cooled in their experiments.

Although in the literature, there exist various theoretical and experimental studies and relations of annulus heat transfer with inner cylinder rotation, and these correlations utilize different criteria to model the effect of rotation and axial flow, we still can't find any empirical Nusselt number relations with Taylor number or Reynolds number that can fully fit the gap ratio, axial ratio, annular diameter ratio, oil velocity and fluid properties of this 2D pump's unique annular mixed mode flow conditions yet. But we can also find some general empirical formulas for turbulent heat transfer in tubes that might suit this situation. Therefore, the purpose of this study is to derive the heat transfer relation of the pump transmission, so as to provide accurate parameters for its thermo-hydraulic model for subsequent analysis of its mechanical efficiency and cooling performance.

## 2. Physical structure

### 2.1 Mechanical structure

This pump is a kind of double-acting piston pump, composing a main piston, two annular balancing pistons, two cam-roller sets, two shaft forks, a cylinder, two end caps, and others. The cylinder has four evenly distributed nozzles around, replacing the traditional oil distribution plate, which are two suction nozzles and two discharge nozzles. And the main piston, which has four channels correspondingly, can be driven to rotate by the right shaft fork in the right end cap.

Two cam-roller sets are set on both sides of the cylinder to make the pistons do reciprocating motion. Two main carriers with four rollers, marked in orange in Fig 2, drive the main piston. And the other two balancing carriers (blue) drive the two balancing pistons. The two pairs of carriers are perpendicular to each other. So the two balancing pistons make homodromy and opposite reciprocation with the main piston synchronously. That's why it's named the

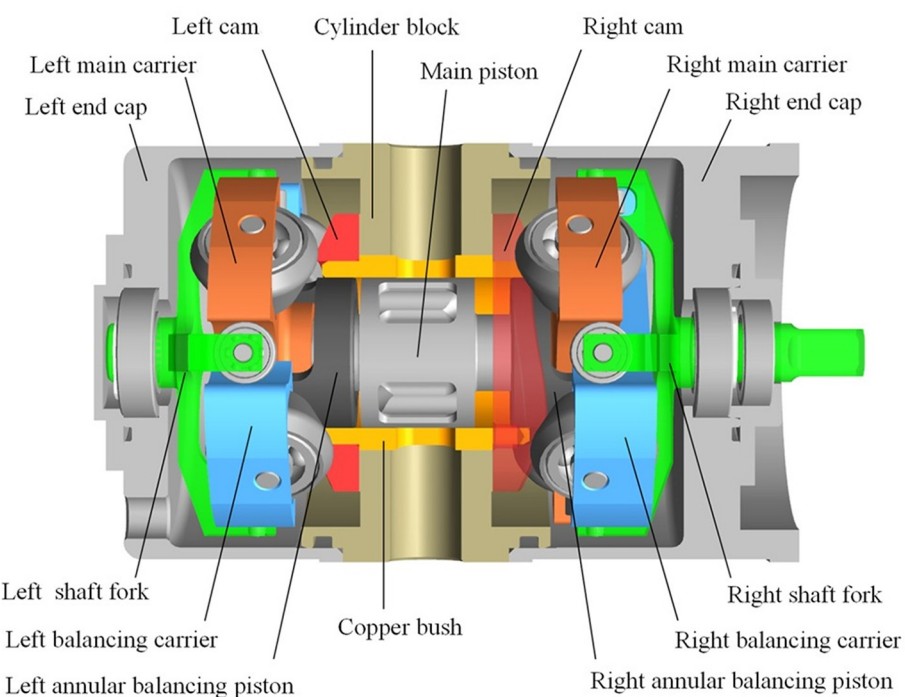

**Fig 2. Schematic configuration of the 2D piston pump with balancing sets.**

two-dimensional (2D) piston pump. The two uniform spatial cams have a 90-degree phase difference in the circumferential direction. The roller assemblies are held tightly against the cams' surfaces. Due to the rotational and reciprocating movement of the pistons, the working volumes between the main piston and balancing pistons, surrounded by the cylinder block, are changed. And this pump can perform four oil suctions and compress processes per cycle.

The right shaft fork delivers 50% input torque to the right balancing carrier through a transmission shaft to the left balancing carrier. For the balancing pistons are connected to their relevant carriers, the balancing pistons and balancing carriers move consistently. And the right fork sends another 50% power to the right main carrier through the main piston to the left main carrier. Likewise, the main piston and main carriers move together. The right shaft fork drives both sides of four roller carriers symmetrically. All the shaft forks, pistons, and roller carriers' rotational speeds are in accordance with the input velocity of rotation. The carrier's tangential velocity $v_1$ is described by Eq 8. Due to the shape of the cam, which is designed according to the law of uniform acceleration and deceleration, their velocities of reciprocating motions are under the rule of it correspondingly. The main carrier and balancing carrier on either side are orthogonal. Thus, their axial velocities are equal in magnitude and opposite in direction ($v_2' = -v_2$), described in Eq 9 and shown in Fig 3. So the balancing sets can not only increase the pump's displacement but also reduce the vibration caused by the reciprocating motion. The balancing sets are highlights compared with the other 2D pumps. But this unique intricate transmission also adds extra contact with oil and makes the thermal-hydraulic model more complex relevantly.

$$v_1 = \omega \cdot r_3 = \frac{2\pi n}{60} \cdot r_3 = 3.14 \times 10^{-3} n \tag{8}$$

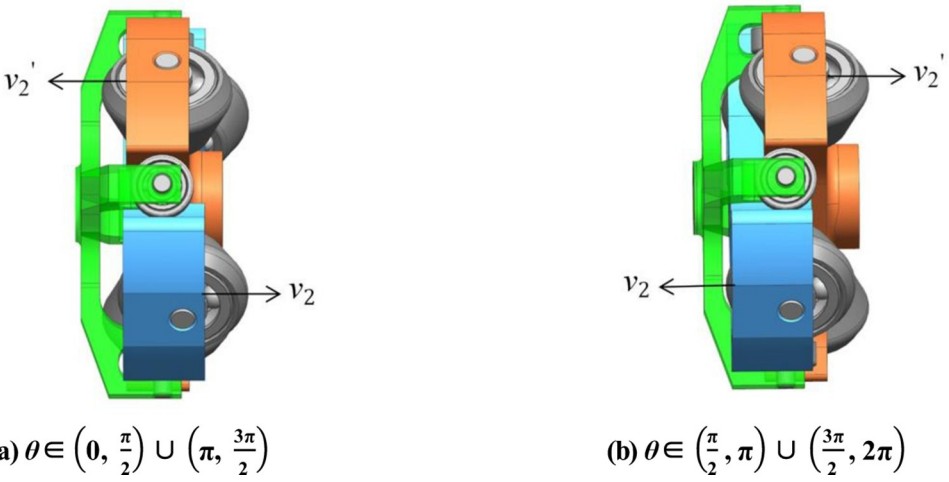

**(a)** $\theta \in \left(0, \dfrac{\pi}{2}\right) \cup \left(\pi, \dfrac{3\pi}{2}\right)$ **(b)** $\theta \in \left(\dfrac{\pi}{2}, \pi\right) \cup \left(\dfrac{3\pi}{2}, 2\pi\right)$

**Fig 3. The axial velocities of the left carriers.** (a) $\theta \in \left(0, \frac{\pi}{2}\right) \cup \left(\pi, \frac{3\pi}{2}\right)$. (b) $\theta \in \left(\frac{\pi}{2}, \pi\right) \cup \left(\frac{3\pi}{2}, 2\pi\right)$..

$$v_2 = \begin{cases} \dfrac{16h}{\pi^2}\omega\theta & , \theta \in \left(0, \dfrac{\pi}{4}\right) \\[2mm] -\dfrac{16h}{\pi^2}\omega\theta + \dfrac{8h}{\pi}\omega, & \theta \in \left(\dfrac{\pi}{4}, \dfrac{3\pi}{4}\right) \\[2mm] \dfrac{16h}{\pi^2}\omega\theta - \dfrac{16h}{\pi}\omega, & \theta \in \left(\dfrac{3\pi}{4}, \dfrac{5\pi}{4}\right) \\[2mm] -\dfrac{16h}{\pi^2}\omega\theta + \dfrac{24h}{\pi}\omega, & \theta \in \left(\dfrac{5\pi}{4}, \dfrac{7\pi}{4}\right) \\[2mm] \dfrac{16h}{\pi^2}\omega\theta - \dfrac{32h}{\pi}\omega, & \theta \in \left(\dfrac{7\pi}{4}, 2\pi\right) \end{cases} \tag{9}$$

From the Fig 4, it can be clearly found that the carrier's reciprocating velocity has nothing to do with time $t$, but only related to the angular velocity of rotation and the maximum stroke h.

The carrier's mean axial velocity:

$$v_{2\text{ave}} = \begin{cases} 1.67 \times 10^{-4}n, & \theta \in (0, \pi/2) \\ -1.67 \times 10^{-4}n, & \theta \in (\pi/2, \pi) \\ 1.67 \times 10^{-4}n, & \theta \in (\pi, 3\pi/2) \\ -1.67 \times 10^{-4}n, & \theta \in (3\pi/2, 2\pi) \end{cases} \tag{10}$$

Ratio of the carrier's tangential velocity to the absolute value of average axial velocity:

$$\frac{v_1}{|v_{2\text{ave}}|} = 18.8 \tag{11}$$

The effective velocity of the carrier, taken as the vector sum of the axial flow velocity and the rotor speed [16] (put forward as in Eq 12), is basic equal to the carrier's reciprocating

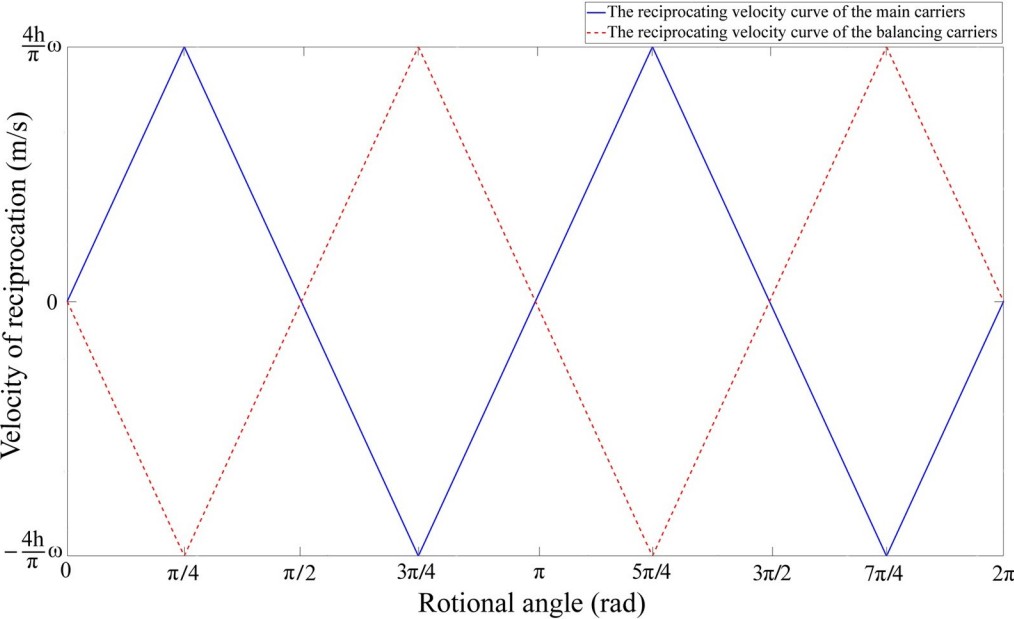

**Fig 4. The reciprocating velocity of the carriers.**

velocity.

$$v_e = \sqrt{{v_1}^2 + {v_{2\text{ave}}}^2} = \sqrt{1.003{v_1}^2} = 3.144 \times 10^{-3}n \tag{12}$$

## 2.2 Flow structure

In this pump's end caps, both sides of the roller carrier assemblies are axisymmetric and eudipleural. The flow structure in such concentric cylinders much depends on its geometric parameters. In the left cap, the left inner cylinder is composed of the left main carrier, the left balancing carrier, and the left shaft fork. Since the diameter of the shaft fork is slightly larger than the carriers' and grooves are existed between them, the inner rotor has bulges and slotted gaps. This inner cylinder rotates to form a unique Taylor mixed mode flow with convex and slits [16, 17]. And the dual carriers make contrary reciprocations to generate two pairs of opposite axial flows. So this thermal-hydraulic flow is a special kind of Taylor-Couette-Poiseuille flow (a three-dimensional flow).

As Fig 5 illustrates, its geometry is basically characterized by the inner radius of the outer cylinder $r_1$ and the radius of the inner rotor $r_3$, as well as the fluid field length $L_{cy}$. Besides, this flow is featured with an array of geometric parameters: annular gap thickness: $e = r_1 - r_3$, gap ratio: $G = e/r_3$, hydraulic diameter: $D_h = 4A_c/P$ [30], axial ratio: $\Gamma = L_{cy}/(r_1 - r_3)$, and diameter ratio: $a = r_1/r_3$. The parameters of the annular flow are shown in Table 1.

This annular flow's dynamic features are formed by the temperature field and velocity field, including the inner annular tangential velocity $v_1$ and axial velocity $v_2$ and $v_2$'. In a Taylor flow, the Taylor number, which is interpreted as the ratio between centrifugal force and viscous force, is more preferred to the Reynolds number. This paper adopts the most widely accepted Taylor number definitions: $T_a = \omega^2 R(D_h/2)^3/\vartheta^2$ [31]. And the critical Taylor number $T_{ac}$ is about 1700, above which the Taylor vortices appear [32]. However, in Gardiner and Sabersky's research, it becomes an unusually high to $10^4$ rather than 1700, for the influence of natural convection [33]. When $T_a/T_{ac}$ exceeds 1300, the annular flow becomes turbulence [34]. In fact,

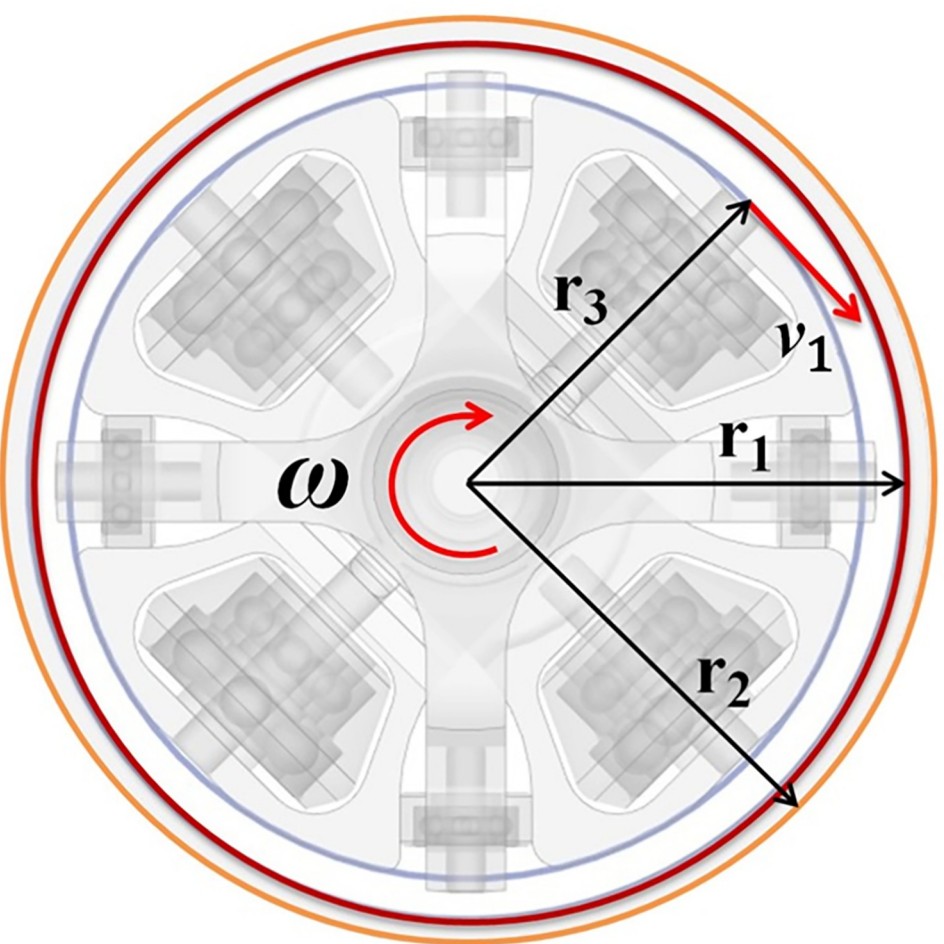

**Fig 5. Geometries of the concentric cylinders.**

as a result of dissimilar geometries of each experiment facility, the critical Taylor numbers are varied a lot with each other with regard to the dimensions of a cylindrical gap. When this inner rotor is at the speed of 1000 rpm, $T_a/T_{ac}$ exceeds 1.2 and the wavy vortex flow appears. At 4000 rpm, $T_a/T_{ac}$ begins to beyond 1300 and the turbulent flow shows up [11]. So with the

**Table 1. Parameters of the flow.**

| Name | Value |
|---|---|
| The inner radius of the outer cylinder $r_1$ | 0.033m |
| The outer radius of the outer cylinder $r_2$ | 0.035m |
| The radius of the inner cylinder $r_3$ | 0.03m |
| The maximum stroke of the piston h | 0.0025m |
| Hydraulic diameter $D_h$ | 0.0061m |
| Fluid field length $L_{cy}$ | 0.013m |
| Annular gap thickness e | 0.003m |
| Gap ratio G | 0.1 |
| Axial ratio $\Gamma$ | 4.33 |
| Annular diameter ratio a | 1.1 |

increased rotation speed of this pump ranging from 1000-6000rpm, the annular flow contains laminar flow with vortices, turbulent flow with vortices, and purely turbulent flow in turn [16, 18].

## 3. Heat transfer modeling and simulation

Due to the churning losses [12] in this pump cap's annular duct, the oil is heated and the heat is transferred to both the inner and outer cylinders meanwhile. So such boundary conditions belong to the heat transfer from both tubes to the annular flow [35]. And this paper focuses on the heat transfer of the outer cylinder rather than the inner one, from the high-temperature oil through its wall to the low-temperature surroundings. The heat transfer outside of the end cap is a natural heat transfer system including the free-convection heat transfer and radiation heat transfer to the air. And the heat transfer from the oil to the wall is a forced-convection heat transfer [36].

### 3.1 Heat transfer modeling

The end cap material is Al2024 aluminum alloy of high thermal conductivity, and the fluid inside is 46 hydraulic oil. The Biot number which compares the relative magnitudes of surface-convection and internal-conduction resistances to heat transfer [36], between the oil and inner wall-face ($B_{i1} = h_1 D_{h1}/k_{cy}$) and between the air and outer wall-face ($B_{i2} = h_2 D_{h2}/k_{cy}$) are all less than 0.1, according to the result of experimental data. So the cylinder's behavior without heat sources ($\dot{E}_g = 0$) could adopt the lumped-heat-capacity method to analyze [30]. Thus, its axial heat conduction can be neglected, and the temperature field is axisymmetric. So the outer annular temperature is approximately uniform throughout its solid ($T_{cy} \approx T_{r2}$).

When the rotational speed is low, the few heat generated by the churning and friction can be fully carried away by the surroundings, and the wall could easily meet the one dimension steady-state conduction. As Fig 6 shows, according to the first law of thermodynamics ($\dot{E}_{in} + \dot{E}_g - \dot{E}_{out} = \dot{E}_{st}$), the input heat flow ($q_{conv1}$) from the oil through the inner wall-face, turning into the cylinder heat flow, is equal to the output heat flow ($q_2$), divided into convection ($q_{conv2}$) and radiation ($q_{rad2}$), through the outer wall-face to the surroundings, and the internal energy of the wall is steady ($\dot{E}_{st} = 0$). While the rotational speed rises high, the heat generated can't be totally taken away, and the wall's temperature keeps changing in the unsteady-state conduction. The internal energy of the wall varies with the temperature gradient. According to Newton's law of cooling $q = A \cdot h \cdot (T_s - T_f)$, including the convection ($q$), the surface area (A), the average convection coefficient ($h$), and the temperature difference of the solid surface and the fluid ($T_s - T_f$), the convection problems reduce to the estimation of the convection coefficient [37].

$$\dot{E}_{in} = q_{conv1} = 2\pi r_1 h_1 L_{cy}(T_{oil} - T_{r1}) \tag{13}$$

$$\dot{E}_{out} = q_2 = q_{conv2} + q_{rad2} \tag{14}$$

$$\dot{E}_{st} = V_{cy}\rho_{cy}c_{cy}\frac{dT_{cy}}{dt} \tag{15}$$

For the pump's surface is covered with thin flat black lacquer, the emissivity $\varepsilon$ is 0.96 [36]. And the radiant exchange of the pump cap can be calculated with Eq 16, where $\sigma$ is the Stefan-Boltzmann constant with the value of 5.669×10$^{-8}$ W /m$^2$·K$^4$.

$$q_{rad2} = 2\pi r_2 L_{cy}\varepsilon \cdot \sigma(T_{r2}^4 - T_{sur}^4) \tag{16}$$

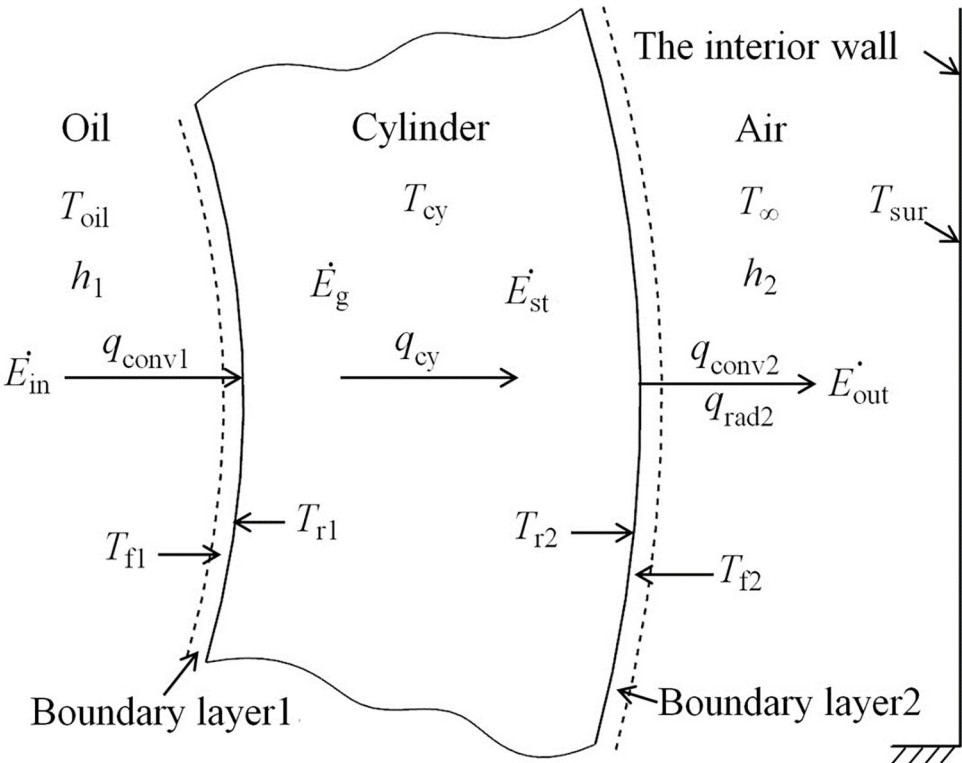

**Fig 6. Overall heat transfer through the cylinder.**

According to the experimental correlation formula for natural convection heat transfer of a horizontal cylinder with a wide range of Rayleigh numbers recommended by Churchill and Chu [38], the Nusselt number of the outer wall-face is given in Relation 17. Thus, the convective heat transfer $h_2$ and the natural convection $q_{conv2}$ can be calculated.

$$N_{u2} = \frac{h_2 D_{h2}}{k_2} = \left\{ 0.6 + \frac{0.387 R_{a2}^{1/6}}{\left[ 1 + \left( \frac{0.559}{P_{r2}} \right)^{9/16} \right]^{8/27}} \right\}^2 \tag{17}$$

$$q_{conv2} = 2\pi r_2 L_{cy} h_2 (T_{r2} - T_\infty) \tag{18}$$

Thus, the convective heat transfer $h_1$ could be expressed in Relation 19. The non-dimensional number $N_{u1}$ can be deduced in the end. It's emphasized that all fluidic properties (oil and air) are evaluated at the film temperatures.

$$h_1 = \frac{r_2 h_2 (T_{r2} - T_\infty) + r_2 \varepsilon \cdot \sigma \left( T_{r2}^4 - T_{sur}^4 \right) + 0.5 (r_2^2 - r_1^2) \rho_{cy} c_{cy} \frac{dT_{cy}}{dt}}{r_1 (T_{oil} - T_{r1})} \tag{19}$$

## 3.2 Heat transfer simulation

To verify the validity of convective heat transfer coefficient $h_1$, a 3D CFD model was implemented with Ansys Transient Thermal software. The geometry of out annulus is consistent with the actual design. As shown in Fig 7(A), the grid is set as an integrated grid to ensure good continuity. And the CFD numerical simulation model parameters, including $h_1$, $T_{oil}$, $T_\infty$,

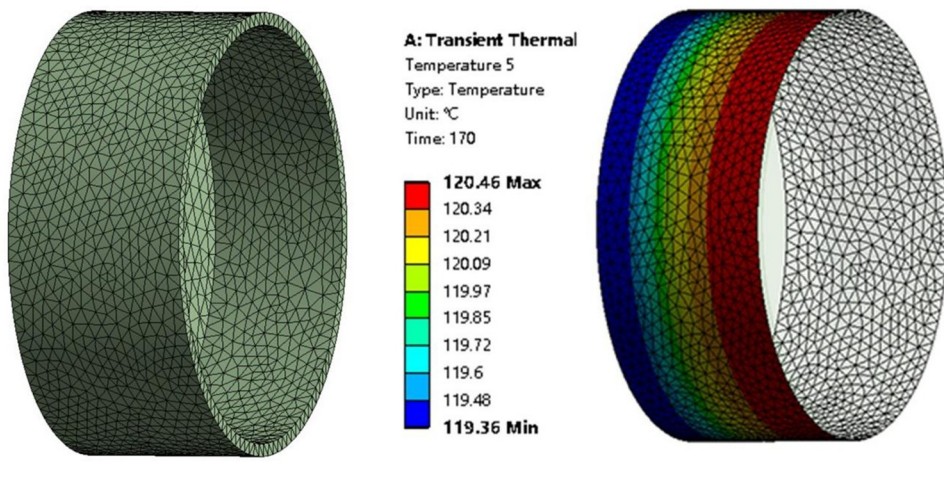

**(a) The grid model**

**(b) The $T_{r2}$ simulation results at 6000 rpm**

**Fig 7. The heat transfer simulation of the outer annulus. (a)** The grid model. **(b)** The $T_{r2}$ simulation results at 6000 rpm.

and $T_{sur}$, were set consistently with the experimental data under the rotational speed from 1000 rpm to 6000 rpm.

The material of the annulus is Al2024. The force convection with tabular data of convection coefficient $h_1$ and oil temperatures is added to the inner face of the annulus. The natural convection with convection coefficient of stagnant air-horizontal cylinder and the radiation with emissivity 0.96 are set on the outer face of the annulus. And the ambient temperature is also in line with the experimental value. Through the transient heat transfer simulation process, the temperature of the out annulus can be directly obtained. And the transient temperature of shell $T_{r2}$ can be calculated, as shown in Fig 7(B).

## 4. Experimental setup

As shown in Fig 8(A), an experimental setup of the heat transfer of this 2D pump was developed. It is composed of a 30kW three-phase asynchronous electric motor driving this pump to

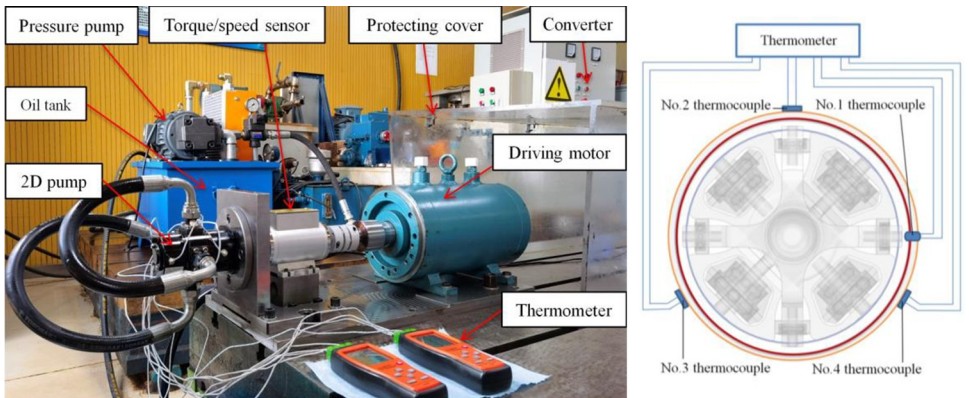

**(a) An overview of the experimental setup**

**(b) Schematic of $T_{oil}$ and $T_{r2}$ measurement system**

**Fig 8. The heat transfer test rig. (a)** An overview of the experimental setup. **(b)** Schematic of $T_{oil}$ and $T_{r2}$ measurement system.

**Table 2. The experimental conditions and devices' details.**

| Description | Value |
|---|---|
| Atmospheric pressure | 101.16 kPa |
| Rotational speed | 1000–6000 rpm |
| Initial environment temperature | 23.5˚C |
| Initial temperature of surroundings | 23.5˚C |
| Smart Sensor AS887 thermometer | Range -200–1372˚C, accuracy ± 0.1% + 0.6˚C, resolution0.1˚C |
| K-type thermocouple | Range -50–300˚C, accuracy ± 1.5% |
| Sanjing SL06 torque/speed sensor | Range 0–10 N·s, accuracy ± 0.1%; Rotational speed range 0–18000 rpm |

the uttermost speed of 10,000 rpm by a converter, a dynamic torque sensor installed along a driving shaft, the 2D pump, one Smart Sensor AS887 4-channel thermometer to measure the left end cap's temperatures, and others. Four K-type thermocouples were set as in Fig 8(B): No.1 was fixed through the cap to measure $T_{oil}$, and the other three were evenly secured around the cap with aluminum tape to test $T_{r2}$. The speed of the rotor can be simply obtained by the torque/speed sensor. Besides, there was another same type of thermometer to monitor the right cap's temperatures, oil temperatures in one suction nozzle and one discharge nozzle for further overall thermal analysis, and a camera placed to shoot videos of the temperature data. The accuracies of the related devices are shown in Table 2. Besides, the temperature errors are less than 0.1˚C.

When the annular structural parameters are constant, the Reynolds number and Taylor number are affected by the fluidic velocity and temperature. In order to produce various velocity and temperature fields, the convective heat transfer coefficients, from lower Reynolds number and Taylor number to higher ones, were derived in this experiment at wide driving speeds of 1000 rpm, 1500 rpm, 2000 rpm, 3000 rpm, 4000 rpm, 5000 rpm, and 6000 rpm. The pump's two suction nozzles were directly connected with two discharge nozzles with two tubes, and it ran without load, separately from the external pressurized pump and oil tank to avoid their interference. To remove air from this system, the end caps and tubes were assembled in 46 hydraulic oil. In this circulatory system, the temperature rose almost due to the churning loss instead of the friction of mechanical parts, for the compression force on the intake oil was almost zero. In this way, the energy losses were mainly caused by heat loss rather than vibration loss. So this experiment could partly verify Huang's results about the churning loss [7, 12] of this pump transmission. In addition, a heat-insulating washer was set between the right end cap and the mounting plate to reduce conduction heat transfer to the test rig. Therefore, the heat transfers of both side end caps are analogous.

## 5. Results and discussion

The test apparatus is used to obtain temperature data to work out the unknown convective heat transfer coefficients of this 2D aero pump's cap and deduce the Nusselt number correlations of this unique Taylor-Couette flow finally. The No.1 thermocouple temperature sensor, set at the intermediate height of the cap's oil level, was totally immersed in the oil. Thus, it can easily obtain $T_{oil}$ ($T_{No.1}$). For the vibration and other causes, the thermocouples stuck on the cylinder wall, were not that easy to accurately and promptly acquire $T_{r2}$. So that's why three sensors were installed and the average of their three measurements was taken as the actual $T_{r2}$ to manipulate ($T_{r2} = \frac{T_{No.2}+T_{No.3}+T_{No.4}}{3}$).

The four thermocouples send signals to the same thermometer to maintain the uniformity of the measure values. In this way, $h_1$ and $h_2$ could be determined relatively accurately. During

the thermal experiment, in order to avoid the influence of the surrounding temperature on the heat loss, each test was carried on under the same room temperature (around 23.5˚C), and the setting rotational speed was reached as soon as possible. The thermometer was manually recorded through videos, and all the captured Celsius temperature data were uniformly converted into Kelvin temperature after all.

## 5.1 The convective heat transfer coefficient

The values of test values and cumulative results, including $T_{oil}$, $T_{r2}$, $R_e$, $T_a$ and $h_1$ are presented in the supporting information. All properties were evaluated at the mean bulk temperatures of the fluids. Thus $h_1$ can be obtained from Formula 19. A log-linear plot of $h_1$ versus $T_a$ is shown in Fig 9, which demonstrates the change of the convective heat transfer coefficient with the Taylor number and rotational speed for the annuli. Seven sets of experiments were conducted, varying the rotational speed from 1000 rpm to 6000 rpm, to estimate the dependence of the heat transfer on the Taylor number. When the rotational speed is below 2000 rpm, the cylinder wall could finally meet the thermal equilibrium (steady-state). While the speed is over 5000 rpm, this annulus is consistent in the unsteady-state heat transfer until the oil boils (saturation temperature about 150˚C) to produce a boiling heat transfer, which is beyond the scope of this study.

The results in Fig 9 demonstrate evidently that $h_1$ rises with log $T_a$ at the same rotational speed. Moreover, the $h_1$ peaks grow more pronounced as the Taylor numbers grow up. Note that the initial curve of 1000 rpm alters differently from others, for the Couette flow is changing into the Taylor vortex flow in this situation. When the speed is over 4000 rpm, the wavy vortex flow begins to develop into the turbulent flow, and $h_1$ appears pronouncedly higher above the threshold. These findings could be explained by a heightening of centrifugal forces and the speed of the vortices with the increased rotation speed [18]. It is obviously observed

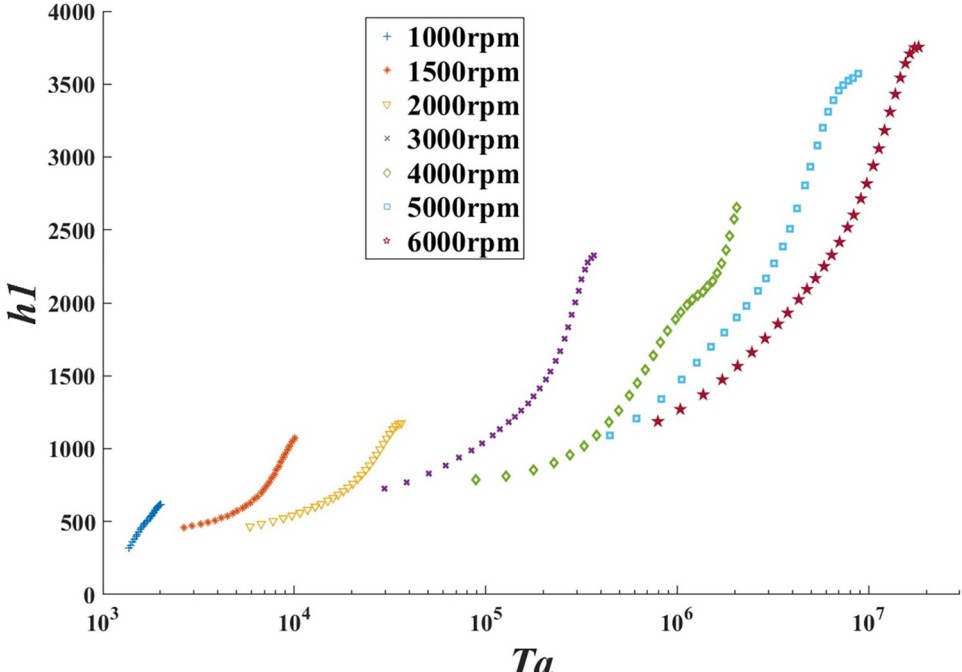

**Fig 9. Experimental convective heat transfer coefficients of different Taylor number and speed.**

that $h_1$ of the lower rotational speed is slightly larger than $h_1$ of the higher one, at the same Taylor number or even lower. Because kinematic viscosity matters Taylor numbers as much as angular velocity, according to the $T_a$ definition. When reaching the same $T_a$, the lower velocity fluid's temperature is higher and its viscosity is smaller, compared with higher speed conditions. The smaller the viscosity of the fluid, the higher the convective heat transfer coefficient [36].

The average temperature of $T_{r2}$ can be calculated and compared with the average temperature of simulation $T_{r2}$, as shown in Fig 10, from 1000 rpm to 6000 rpm. Moreover, Table 3 demonstrates the simulation results can represent the experimental data within ± 2% relative deviation, and the simulation results agree well with the experimental values of $T_{r2}$. So it can be concluded that the annular heat transfer model, the experimental measurement data, and the deduced convective heat transfer coefficient $h_1$ are accurate.

## 5.2 The Nusselt number

For the forced-convection heat transfer of tube flow, the most widely proposed estimation of the Nusselt number is empirically formulated in the form of experimental correlations under various flow structures:

$$N_u = CR_e{}^m P_r{}^n \tag{20}$$

where the heat transfer data depend on the Reynolds number and Prandtl number [36]. The heat transfer coefficients for annular turbulent flow are different from the tubes', for the effect of the inner cylinder on the velocity profile which is taken into account by the annular diameter ratio [35].

And for the Taylor turbulent flow heat transfer, most authors recommend the following relation:

$$N_u = CT_a{}^m P_r{}^n \tag{21}$$

where the Nusselt number is appreciably in conjunction with $T_a$ rather than $R_e$, and the heat transfer significantly depends on the Taylor number and Prandtl number [18].

In Relations 20 and 21, the constant values C, m, and n are to be determined from the test data [39], depending a priori on the influence of experimental conditions, including aspect ratio of the cylindrical gap, axial flow rate, rotation velocity or effective velocity, etc. [16]

The Taylor number increases rapidly with the raise of the annular flow's velocity and temperature, while the oil viscosities decrease with temperature increment [36]. Meanwhile, the Prandtl number exhibits a downward trend when oil temperature increases (46 hydraulic oil & below 100°C). Thus, the influence of the Prandtl number is negligible compared with the Taylor number on the Taylor flow and turbulent flow heat transfer.

A log-linear plot of $N_{u1}$ versus $T_a$ is shown in Fig 11, which demonstrates the development of the Nusselt number with the Taylor number and rotational speed for the annular flow. When the Taylor number is over $2.5 \times 10^5$, the curves in Fig 11 suggest the wavy vortex flow and turbulent flow may occur in turn and hence the $N_{u1}$ peaks grow more significantly. And the lower rotation speed's $N_{u1}$ is slightly larger than the higher speed one with the same $T_a$.

The least-square fitting method was utilized to determine the approximate values of constant C, exponent m, and n in computation. Therefore, two new correlations are suggested, where n is equal to zero, compatible with these cumulative results for such Taylor-Couette-Poiseuille flow conditions. They are helpful to determine high Taylor number heat transfer, especially for 2D pumps. Furthermore, their curves are discontinuous, for the lower rotation

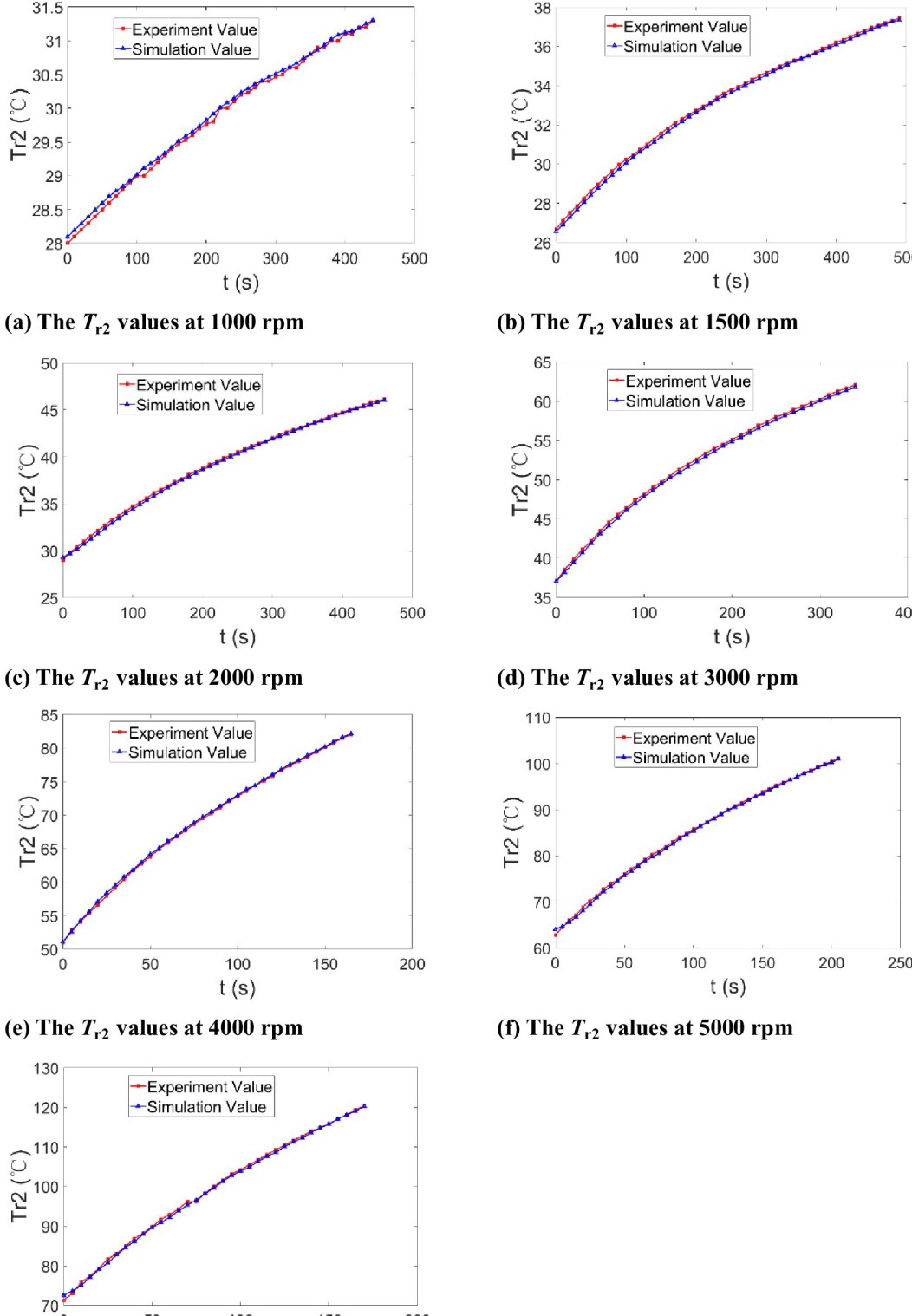

**(a) The $T_{r2}$ values at 1000 rpm**

**(b) The $T_{r2}$ values at 1500 rpm**

**(c) The $T_{r2}$ values at 2000 rpm**

**(d) The $T_{r2}$ values at 3000 rpm**

**(e) The $T_{r2}$ values at 4000 rpm**

**(f) The $T_{r2}$ values at 5000 rpm**

**(g) The $T_{r2}$ values at 6000 rpm**

**Fig 10. The heat transfer simulation of the outer annulus. (a)** The $T_{r2}$ values at 1000 rpm. **(b)** The $T_{r2}$ values at 1500 rpm. **(c)** The $T_{r2}$ values at 2000 rpm. **(d)** The $T_{r2}$ values at 3000 rpm. **(e)** The $T_{r2}$ values at 4000 rpm. **(f)** The $T_{r2}$ values at 5000 rpm. **(g)** The $T_{r2}$ values at 6000 rpm.

**Table 3. Deviations of simulation values to experimental values of $T_{r2}$.**

| Speed(rpm) | 1000 | 1500 | 2000 | 3000 | 4000 | 5000 | 6000 |
|---|---|---|---|---|---|---|---|
| Relative deviation range | -0.14%~0.39% | -0.80%~-0.09% | -1.11%~1.00% | -1.08%~0.06% | -0.55%~0.89% | -1.16%~1.91% | -1.18%~1.69% |
| Mean deviation | 0.17% | -0.40% | -0.47% | -0.63% | 0.29% | -0.32% | -0.30% |

speed's $N_{u1}$ is slightly larger than the higher speed one in the same $T_a$.

$$1300 < T_a < 2.5 \times 10^5 : \quad N_u = 3.9564 T_a^{0.2202} \tag{22}$$

$$2.5 \times 10^5 < T_a < 1.8 \times 10^7 : \quad N_u = 1.7199 T_a^{0.2707} \tag{23}$$

Fig 12 shows a log-linear plot of the ratio of the suggested Nusselt numbers to the experimental results as a function of $T_a$. The Nusselt numbers calculated by the Eqs 22 and 23 are compared with the experimental Nusselt numbers, and the results range from 0.7778 to 1.3383.

Furthermore, the deviations of suggested Nusselt numbers from Eqs 22 and 23 to the experimental Nusselt numbers are specialized exhibited in Table 4. The two new suggested relations can represent the data within the utmost 15% mean deviation.

## 5.3 Discussion

The heat transfer varies significantly with both Reynolds number and Prandtl number [16]. Most of these relations are forms of the annular diameter ratio, Prandtl number, and Reynolds number, corresponding with the Dittus-Boelter type. Therefore, this Taylor flow's experimental results are validated against the Dittus-Boelter and McAdams correlations suggested in Dirker and Meyer's literature [25, 39–41].

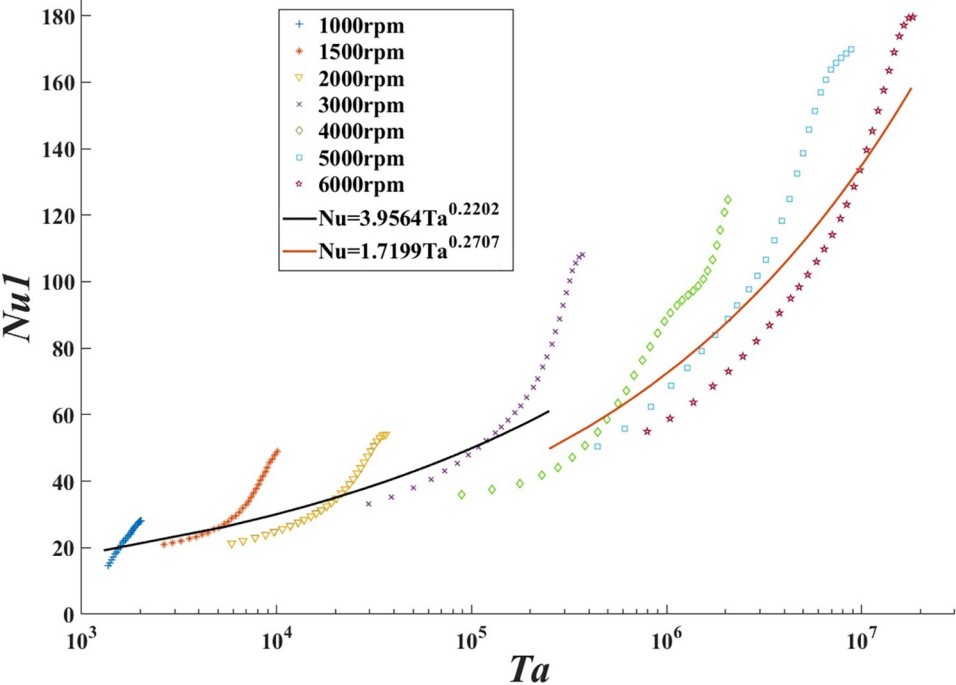

**Fig 11. Experimental and suggested Nusselt number of different Taylor number and speed.**

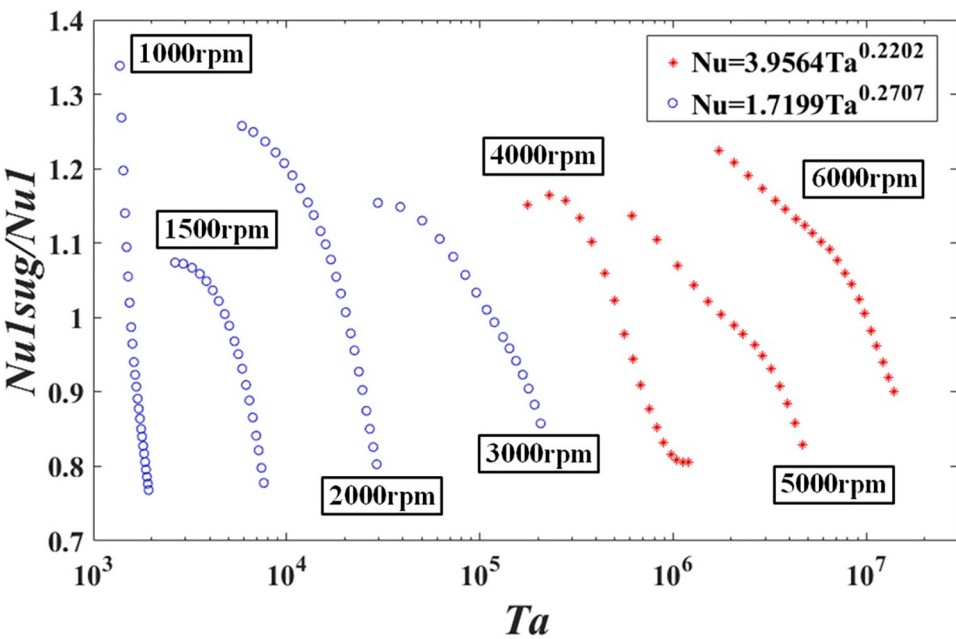

**Fig 12. Ratio of the suggested Nusselt number to experimental results from Eqs 22 and 23 as the function of the Taylor number.**

A log-linear plot of $N_{u1}/P_{r1}^{1/3}$ versus $R_e$ (up to $2.432 \times 10^4$) is shown in Fig 13, which presents the development of the Nusselt number with the Reynolds number and rotational speed for this annular flow. From 1000 rpm to 3000 rpm and $R_e$ up to 2934, the experimental results are in somehow good agreement with the two empirical equations. However, the experimental results vary a lot from these reference relations, when the speeds are beyond 4000 rpm. The experimental results do not give good agreement with the comparable ones and exhibit an even more separation tendency with the increasing of $R_e$.

And the deviations of experimental Nusselt numbers to the referenced Nusselt numbers from Eqs 1 and 2 are demonstrated in Fig 14 and Table 5 in detail.

Fig 14 shows a log-linear plot of ratio of the experimental Nusselt numbers to the reference Nusselt numbers from Eqs 1 and 2 as a function of $R_e$. The Nusselt correlations, derived from the experiments, are compared with Dittus-Boelter and McAdams reference relations simultaneously. In addition, the experimental Nusselt numbers' relative deviations from Dittus-Boelter and McAdams equations are particularly illustrated in Table 5. With $R_e$ from 332.1 to 555.8, the relative deviations of the experimental results from the Dittus-Boelter equation show more agreement than the McAdams equation, below 1500 rpm. And with $R_e$ from 484.9 to 2934, the relative deviations of the experimental results from the McAdams equation give slightly better heat transfer results than the Dittus-Boelter equation at 3000 rpm. While the speeds exceed 4000rpm and Reynolds numbers beyond 2664, the deviations gradually increase. Due to the special annular heat transfer characteristics of the two-dimensional pump

**Table 4. Deviations of Nusselt numbers from Eqs 22 and 23 to the experimental Nusselt numbers.**

| Speed(rpm) | 1000 | 1500 | 2000 | 3000 | 4000 | 5000 | 6000 |
|---|---|---|---|---|---|---|---|
| Relative deviation range | -23.18%~33.83% | -22.22%~7.39% | -19.72%~25.75% | -14.23%~15.41% | -19.46%~15.17% | -17.16%~13.62% | -9.88%~22.37% |
| Mean deviation | 14.55% | 8.64% | 13.47% | 8.04% | 12.71% | 7.27% | 9.43% |

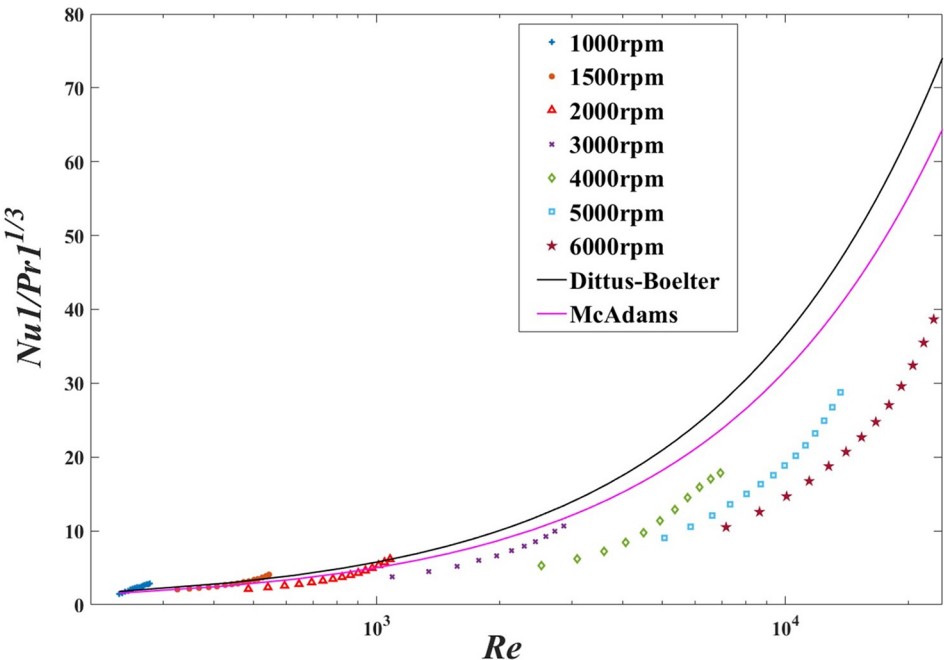

**Fig 13. Experimental and reference Nusselt number of different Reynolds number and speed.**

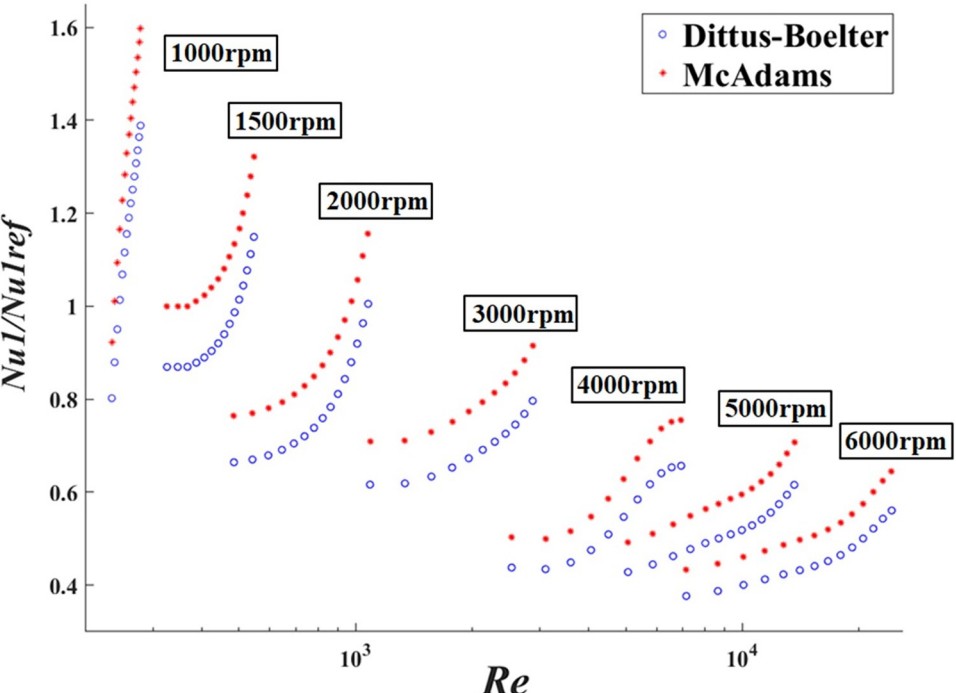

**Fig 14. Ratio of the experimental results and reference Nusselt number from Eqs 1 and 2 as the function of the Reynolds number.**

**Table 5. Relative deviation range of test Nusselt numbers to the Nusselt numbers from Eqs 1, 2.**

| Ref. Eq | 1000 rpm | 1500 rpm | 2000 rpm | 3000 rpm | 4000 rpm | 5000 rpm | 6000 rpm |
|---|---|---|---|---|---|---|---|
| Eq 1 | -19.78%~38.87% | -13.06%~14.92% | -33.57%~0.52% | -38.37%~-20.34% | -56.21%~-34.32% | -57.18%~-38.42% | -62.34%~-43.94% |
| Eq 2 | -7.75%~59.7% | -0.01%~32.16% | -23.6%~15.6% | -29.12%~-8.39% | -49.64%~-24.47% | -50.76%~-29.19% | -56.69%~-35.53% |

at high speed and large Reynolds number, the experimental results have a large deviation from the reference ones.

The deviations become even larger as $R_e$ increasing. The phenomenon can be explained that high shearing cavitation appears with the speed up, and the bubbles in the boundary layer will make the convective heat transfer coefficient decline. This annular flow belongs to high shear flow, due to the shear rates ($D = v_e/e = 1.048n$) ranging from 1048/s at 1000 rpm to 6288/s at 6000 rpm. The strong shear flow correlates with the generation of small bubbles from cavitation [42].

The small bubble cavitation occurs with the Taylor vortex and turbulent vortex when the local minimum pressure drops below the oil vapor pressure [43] and the rotational speed n ≥ 3000 rpm [42]. And the higher the inner annular rotation speed, the more shear stress, and bubble cavitation generated. The bubbles adhering to the outer annuli will make a negative impact on the heat transfer from the two-phase flow to the cap.

Fig 15 illustrates three example photos of the annular two-phase flow in the pump cap. Through an observation hole on the cap, the bubbles could be clearly found and photographed when the rotational speed $n$ beyond 2000 rpm. Fig 15(A) is the result at $n$ = 2000 rpm and $T_{oil}$ = 46.7˚C. We can see a few bubbles randomly distributed in the oil. Fig 15(B) is the result at $n$ = 4000 rpm and $T_{oil}$ = 82.8˚C. It is clearly observed that many small bubbles were formed in the annulus. Fig 15(C) shows the result of $n$ = 6000 rpm and $T_{oil}$ = 106.1˚C, and a large number of small bubbles had been generated inside the rotating flow. So increase in the rotational speed and oil temperature can promote the inception and development of shear cavitation, and the mean volumetric void fraction exhibits an upward trend [44, 45].

In addition, as the solubility of the gas in oil decreases with the increasing temperature, releasing of the dissolved gas also has a negative influence on the convective heat transfer coefficient. The dynamics of gas-liquid cavitation flow structures interact closely with the velocity and temperature dynamics. Thence, the thermal conductivity of oil deviates more from the

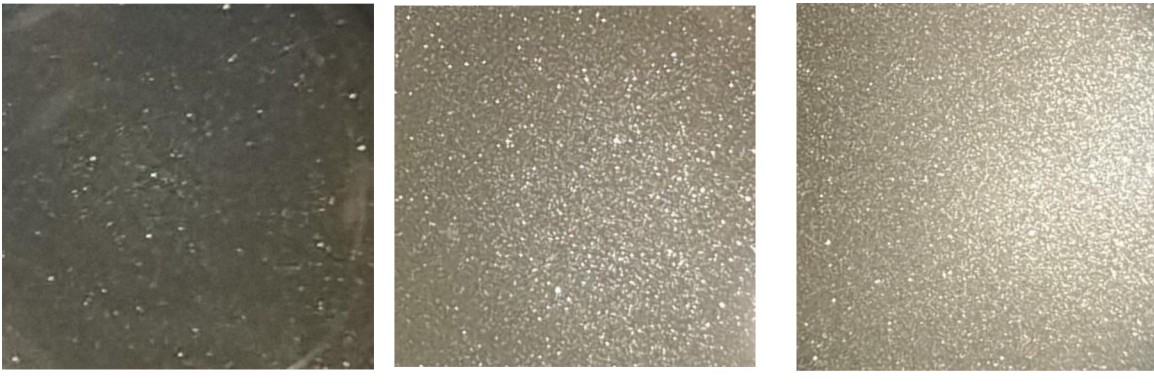

**(a) $n$ = 2000 rpm, $T_{oil}$ = 46.7℃**    **(b) $n$ = 4000 rpm, $T_{oil}$ = 82.8℃**    **(c) $n$ = 6000 rpm, $T_{oil}$ = 106.1℃**

**Fig 15. Photographs of the annular two-phase flow.** (a) $n$ = 2000 rpm, $T_{oil}$ = 46.7˚C. (b) $n$ = 4000 rpm, $T_{oil}$ = 82.8˚C. (c) $n$ = 6000 rpm, $T_{oil}$ = 106.1˚C.

**Table 6. Mean deviation of experimental Nusselt number to the Dittus-Boelter relation and correction factor α at different speed.**

| Speed(rpm) | 1000 | 1500 | 2000 | 3000 | 4000 | 5000 | 6000 |
|---|---|---|---|---|---|---|---|
| Mean deviation | 15.49% | 3.42% | -21.11% | -30.63% | -45.41% | -48.27% | -54.31% |
| Correction factor $\alpha$ | 1.155 | 1.034 | 0.789 | 0.694 | 0.546 | 0.517 | 0.457 |

reference value, when the rise of rotational speed and oil temperature eventually aggravate the development of horizontal annular-dispersed two-phase flow [46, 47]. And this is also the main reason to explain that the lower rotation speed's $N_{u1}$ and $h_1$ are slightly larger than the higher speed ones at the same $T_a$ or $R_e$.

It's assumed that the effect of oil cavitation to convective heat transfer from the oil to the contact surfaces in the pump cap is consistent. A modified Dittus-Boelter relation $N_u = 0.023\alpha R_e^{0.8} P_r^{0.3}$ is introduced to cope with the unknown convective heat transfer of the vortex flows in the pump transmission, from 1000 rpm to 6000 rpm. The factor α is set to correct the deviation caused by oil cavitation. And $\alpha$ is determined through experimental Nusselt number to the Dittus-Boelter relation, as shown in Table 6. Deduction of the correct convective heat transfer coefficient is a necessary task to calculate the thermal status of the pump transmission accurately.

## 6. Conclusions

In this paper, a theoretical and experimental study on heat transfer analysis of the Taylor flow in a 2D piston pump is presented. Based upon the transient thermal simulation results mutually validated with the experimental ones, the following conclusions can be obtained:

1. At the same rotational speed, $N_{u1}$ increases with the rise of Taylor number. As speeding up, the peaks of $N_{u1}$ curves grow more pronounced. $N_{u1}$ at lower rotation speed is slightly larger than the higher speed ones at the same Taylor number because the lower velocity fluid has higher temperature and lower viscosity.

2. The Dittus-Boelter and McAdams relations are not applicable to the pump above 3000rpm, because the void fraction increases with the rotational speed and oil temperature, and the increasing bubbles reduce the convective heat transfer. Therefore, two new suggested heat transfer Relations 22 and 23, representing the data within 15% mean deviation, are determined to evaluate the accurate heat transfer of the pump transmission.

3. A modified Dittus-Boelter relation is introduced to deal with the convective heat transfer of the turbulent flows in the pump transmission. Through comparative analysis, the correction factor α is determined as 1.155 at 1000 rpm, 1.034 at 1500 rpm, 0.789 at 2000 rpm, 0.694 at 3000 rpm, 0.546 at 4000 rpm, 0.517 at 5000 rpm, and 0.457 at 6000 rpm.

The conclusions in this paper can provide a foundation for the accurate thermal status analysis of the 2D pump transmission.

## Supporting information

**S1 Table. The experimental and simulation values with calculated results at 1000 rpm.** (DOCX)

**S2 Table. The experimental and simulation values with calculated results at 1500 rpm.** (DOCX)

**S3 Table. The experimental and simulation values with calculated results at 2000 rpm.** (DOCX)

**S4 Table. The experimental and simulation values with calculated results at 3000 rpm.** (DOCX)

**S5 Table. The experimental and simulation values with calculated results at 4000 rpm.** (DOCX)

**S6 Table. The experimental and simulation values with calculated results at 5000 rpm.** (DOCX)

**S7 Table. The experimental and simulation values with calculated results at 6000 rpm.** (DOCX)

## Acknowledgments

The authors would like to thank Doctor Chuan Ding for the experimental discussion and Doctor Yong Chen for the simulation suggestion.

## Author Contributions

**Conceptualization:** Jian Ruan.

**Data curation:** Liang Chang.

**Formal analysis:** Jian Ruan.

**Funding acquisition:** Liang Chang, Wenang Jia, Sheng Li.

**Validation:** Liang Chang, Zhiwei Li.

**Writing – original draft:** Liang Chang.

**Writing – review & editing:** Wenang Jia.

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
