## [Decision Letter · Decision Letter 0]

15 Jul 2022

PONE-D-22-16038Convective heat transfer of the Taylor flow in a two-dimensional piston pumpPLOS ONE

Dear Dr. Ruan,

Thank you for submitting your manuscript to PLOS ONE. After careful consideration, we feel that it has merit but does not fully meet PLOS ONE’s publication criteria as it currently stands. Therefore, we invite you to submit a revised version of the manuscript that addresses the points raised during the review process.

We look forward to receiving your revised manuscript.

Kind regards,

Yanping Yuan

Academic Editor

PLOS ONE

Journal Requirements:

A clean copy of the edited manuscript (uploaded as the new *manuscript* file).

Reviewers' comments:

Reviewer's Responses to Questions

**Comments to the Author**

1. Is the manuscript technically sound, and do the data support the conclusions?

Reviewer #1: Yes

Reviewer #2: Yes

2. Has the statistical analysis been performed appropriately and rigorously? 

Reviewer #1: Yes

Reviewer #2: Yes

3. Have the authors made all data underlying the findings in their manuscript fully available?

Reviewer #1: Yes

Reviewer #2: No

4. Is the manuscript presented in an intelligible fashion and written in standard English?

Reviewer #1: Yes

Reviewer #2: Yes

5. Review Comments to the Author

Reviewer #1: I recommend the manuscript for publications after these major revisions

1. The manuscript's English language should be improved by the authors.

2. Introduction needs to enrich the readers with recent papers related to your work, which gives the reader a clear visual of the gap that those studies have not address and covered by this study.

3. The results are not enough to defend the importance of work, please add more quantitative outcomes.

4. The reviewer couldn't see the significant objectives of the current work. It would be better if you add a separate section with Key objectives heading.

5. Why this study is more important and how it could contribute to solving the relevant problem.

6. The methodology part doesn't clear; it may not catch the attraction. Try to make it more concise and briefer.

Reviewer #2: The manuscript presents a series of experimental and numerical studies of flow in an annulus, which the authors attribute to a Taylor-Couette like system. They attempt to study this system to model it as a lumped element in a heat conduction problem. This approach is sound, and the authors provide a series of simulations and experiments for this. Overall the analysis is well done and the correlations are explained. I would just like the authors to be more careful with their English as there is a number of typos like "Taylore" or "existing literatures" in the manuscript which need to be addressed.

6. PLOS authors have the option to publish the peer review history of their article (what does this mean?). If published, this will include your full peer review and any attached files.

Reviewer #1: No

Reviewer #2: No

---

## [Author Response · Author response to Decision Letter 0]

30 Aug 2022

Revieser:1

Comment 1 The manuscript's English language should be improved by the authors.

Response Thank you very much for your recognition of our research and for this valuable suggestion. We have edited and corrected the language usage, spelling, grammar, and forma errors of the entire manuscript thoroughly.

Comment 2 Introduction needs to enrich the readers with recent papers related to your work, which gives the reader a clear visual of the gap that those studies have not address and covered by this study.

Response Thank you very much for your suggestion, we have added more recent literature related to this manuscript. See the introduction section of the manuscript for details.

Comment 3 The results are not enough to defend the importance of work, please add more quantitative outcomes.

Response Thank you very much for your suggestion. We have added a modified relation to the gas-liquid two-phase flow heat transfer in the pump transmission for the accurate calculation of its thermal status, as shown below. 

Comment 4 The reviewer couldn't see the significant objectives of the current work. It would be better if you add a separate section with Key objectives heading.

Response Thank you very much for your suggestion, we have added some background of this study. See the introduction section of the manuscript for details. The convective heat transfer coefficients deduced in this work is a necessary preparation for the accurate calculation of the pump’s heat loss. Besides, the results show that the traditional empirical relations are not suitable when the speed is above 3000rpm.

Comment 5 Why this study is more important and how it could contribute to solving the relevant problem.

Response Thank you very much for your suggestion, we have made the abstract, introduction and conclusions changed. We hope you can understand that the deducing of the convective heat transfer coefficient is a fundamental work. The results will be set in the whole pump’s simulation model as the key parameters.

Comment 6 The methodology part doesn't clear; it may not catch the attraction. Try to make it more concise and briefer.

Response Thank you very much for your suggestion, we have added some details to catch the attraction, shortened the unnecessary discussion, and made some necessary changes to the abstract, introduction, results, and conclusions.

Revieser:2

Comment 1 The manuscript presents a series of experimental and numerical studies of flow in an annulus, which the authors attribute to a Taylor-Couette like system. They attempt to study this system to model it as a lumped element in a heat conduction problem. This approach is sound, and the authors provide a series of simulations and experiments for this. Overall the analysis is well done and the correlations are explained. I would just like the authors to be more careful with their English as there is a number of typos like "Taylore" or "existing literatures" in the manuscript which need to be addressed.

Response Thank you very much for your affirmation of our research and for pointing out its shortcomings. We have corrected the typos, syntax and forma errors of the whole manuscript.

---

## [Decision Letter · Decision Letter 1]

27 Sep 2022

Convective heat transfer of the Taylor flow in a two-dimensional piston pump

PONE-D-22-16038R1

Dear Dr. Ruan,

We’re pleased to inform you that your manuscript has been judged scientifically suitable for publication and will be formally accepted for publication once it meets all outstanding technical requirements.

Kind regards,

Yanping Yuan

Academic Editor

PLOS ONE

Additional Editor Comments (optional):

Reviewers' comments:

Reviewer's Responses to Questions

**Comments to the Author**

1. If the authors have adequately addressed your comments raised in a previous round of review and you feel that this manuscript is now acceptable for publication, you may indicate that here to bypass the “Comments to the Author” section, enter your conflict of interest statement in the “Confidential to Editor” section, and submit your "Accept" recommendation.

Reviewer #1: All comments have been addressed

Reviewer #2: All comments have been addressed

2. Is the manuscript technically sound, and do the data support the conclusions?

Reviewer #1: Yes

Reviewer #2: Yes

3. Has the statistical analysis been performed appropriately and rigorously? 

Reviewer #1: N/A

Reviewer #2: Yes

4. Have the authors made all data underlying the findings in their manuscript fully available?

Reviewer #1: Yes

Reviewer #2: Yes

5. Is the manuscript presented in an intelligible fashion and written in standard English?

Reviewer #1: Yes

Reviewer #2: Yes

6. Review Comments to the Author

Reviewer #1: The authors have addressed all of my comments

The authors have addressed all of my comments

The authors have addressed all of my comments

Reviewer #2: (No Response)

7. PLOS authors have the option to publish the peer review history of their article (what does this mean?). If published, this will include your full peer review and any attached files.

Reviewer #1: No

Reviewer #2: No

---

## [Editor Report · Acceptance letter]

30 Sep 2022

PONE-D-22-16038R1 

Convective heat transfer of the Taylor flow in a two-dimensional piston pump 

Dear Dr. Ruan:

I'm pleased to inform you that your manuscript has been deemed suitable for publication in PLOS ONE. Congratulations! Your manuscript is now with our production department. 

Kind regards, 

on behalf of

Prof. Yanping Yuan 

Academic Editor

PLOS ONE